# Reentrant Behavior of the Density vs. Temperature of Indium Islands on GaAs(111)A

**DOI:** 10.3390/nano10081512

**Published:** 2020-07-31

**Authors:** Artur Tuktamyshev, Alexey Fedorov, Sergio Bietti, Shiro Tsukamoto, Roberto Bergamaschini, Francesco Montalenti, Stefano Sanguinetti

**Affiliations:** 1Laboratory for Nanostructure Epitaxy and Spintronics on Silicon (L-NESS) and Material Science Department of the University of Milano-Bicocca, via. R. Cozzi 55, 20125 Milano, Italy; sergio.bietti@unimib.it (S.B.); shiro.tsukamoto@unimib.it (S.T.); roberto.bergamaschini@unimib.it (R.B.); francesco.montalenti@unimib.it (F.M.); stefano.sanguinetti@unimib.it (S.S.); 2Laboratory for Nanostructure Epitaxy and Spintronics on Silicon (L-NESS) and Istituto di Fotonica e Nanotecnologie - National Research Council (CNR-IFN), via F. Anzani 42, 22100 Como, Italy; alexey.fedorov@polimi.it

**Keywords:** droplet epitaxy, GaAs(111)A, indium islands, RHEED, liquid-solid transition

## Abstract

We show that the density of indium islands on GaAs(111)A substrates have a non-monotonic, reentrant behavior as a function of the indium deposition temperature. The expected increase in the density with decreasing temperature, indeed, is observed only down to 160 °C, where the indium islands undertake the expected liquid-to-solid phase transition. Further decreasing the temperature causes a sizable reduction of the island density. An additional reentrant increasing behavior is observed below 80 °C. We attribute the above complex behavior to the liquid–solid phase transition and to the complex island–island interaction which takes place between crystalline islands in the presence of strain. Indium solid islands grown at temperatures below 160 °C have a face-centered cubic crystal structure.

## 1. Introduction

Droplet epitaxy (DE) is a well-established method for producing quantum dots (QDs) with an independent control of nanostructure size and its density in the wide range [1,2]. Using DE, it is possible to fabricate low-density QDs for quantum emitters [1,2,3,4] or high-density nanostructures for laser and photodetector devices [5,6,7].

The common strain-driven Stranski–Krastanov (SK) growth technique for QD fabrication, like the prototypical InAs/GaAs system [8], is not able produce QDs self-assembly on (111) surfaces due to the low threshold energy for compressive strain relaxation in epitaxial layers by the insertion of misfit dislocations at the substrate–epilayer interface [9,10]. Nevertheless, by DE it is possible to realize highly symmetric QDs grown on (111) surfaces with natural C${}_{3v}$ symmetry, which hold high enough symmetry to prevent fine structure splitting (FSS) of the exciton state as recently shown in lattice-matched GaAs/AlGaAs QDs systems at a wavelength in the 750 to 800 nm range [3,4,11,12].

To shift emission wavelength up to telecommunication bands (1.31–1.55 μm), it becomes necessary to adapt the heterostructure composition to allow for the emission in the required energy range. One possible way to do this is to fabricate InAs QDs embedded in InGa(Al)As layers, pseudomorphically grown on InP substrates [13,14] or metamorphically grown on GaAs substrates [15,16,17,18].

An additional issue, related to the epitaxial growth on a singular (111) surface, is the formation of large pyramidal hillocks, nucleated by stacking faults [19]. To create well-defined QDs, a flat surface should be obtained. Optimal growth conditions for Ga(Al)As layers on a singular GaAs(111)A surface are low growth rate and high V/III flux ratio [20], so the growth of QD embedding in a planar cavity with the thick distributed Bragg reflectors (DBRs) becomes problematic on the singular (111) surface. These critical requirements can be mitigated using a vicinal (111) surface, in which the growth conditions (high growth rate and low V/III ratio) are similar to those for a GaAs(001) surface thanks to the presence of preferential nucleation sites at the step edges. Ga droplet nucleation on the vicinal GaAs(111)A substrate was already studied and a temperature activated dimensionality crossover at ~400 °C in the nucleation of QDs was observed [21], which has the effect of decreasing the droplet density activation energy at high temperatures.

In this work, we present the investigation of indium islands self-assembly (the first step of InAs QD fabrication by DE technique) on GaAs(111)A substrates, both nominal and miscut, in a solid source molecular beam epitaxy (MBE) system. We find that the indium island density shows a complex, reentrant behavior, determined by the interplay of the nucleation dynamics, liquid–solid phase transition, and crystalline phase-activated island–island mass transfer effects. Reflection of high energy electron diffraction (RHEED) analysis shows that indium solid islands grown at temperatures below 160 °C have a face-centered cubic (FCC) crystal structure.

## 2. Experimental Methods

Indium islands were self-assembled on semi-insulating GaAs(111)A substrates. On-axis and misoriented (111)A with 2° miscut towards $\left(\bar{1}\;\bar{1}\;2\right)$ were used. After an oxide desorption at 590 °C for 5 min, a 130 nm GaAs buffer layer was deposited to smooth the surface. The substrate temperature T was varied from 30 to 395 °C. T was measured by thermocouple, situated between the substrate heater and the sample holder. The temperature was calibrated by the desorption of native oxide and appearance of (2 × 2) reconstruction at 580 °C and by melting a piece of indium, attached to sample holder, at 156.6 °C. The total amount of indium supplied for the island formation was 2 monolayers (ML) (here and below 1 ML is defined as 6.26 × 10${}^{14}$ atoms/cm${}^{2}$, which is the site-number density of the unreconstructed GaAs(001) surface), deposited with a growth rate of 0.04 ML/s. For one sample 1 ML of In was deposited at 80 °C, to check the influence of a deposited indium amount on the island density. During the indium deposition the background pressure was below 3 × 10${}^{-9}$ Torr. The supply of indium without As${}_{4}$ enabled appearance of indium liquid droplets or indium solid islands on the surface, depending on the deposition temperature. Every growth experiment was monitored in situ by RHEED. During the growth of GaAs buffer layer only (2 × 2) reconstruction was observed. The morphological characterization of the samples was performed ex situ by an atomic force microscope (AFM) in tapping mode, using tips capable of a resolution of approximately 2 nm.

## 3. Results and Discussion

Using the DE technique, it is possible to control the droplet density and in turn the QD density by varying the droplet deposition temperature and flux rate [1,2]. Figure 1 shows AFM images of indium islands obtained on vicinal GaAs(111)A at different deposition temperature and the same In flux rate. From the nucleation theory of Venables [22,23], the density of stable clusters exponentially depends on the temperature: with decreasing the deposition temperature the density of stable clusters is increasing. The density dependence of Ga droplets and GaAs QDs self-assembled on GaAs(001) [24], on singular GaAs(111)A [25,26], and on vicinal GaAs(111)A [21], as well as In droplets and InAs QDs on GaAs(001) [27] and on InP(001) [28] satisfy the behavior described by the nucleation theory.

As it is clear from Figure 2, the density of indium islands self-assembled on GaAs(111)A displays a complex dependence on the deposition temperature. In the reported 30–395 °C temperature range, indeed, deviations from the monotonic increase of density with decreasing temperature predicted by nucleation theory [22,23] is quite evident. The same behavior in the range of 100 to 200 °C is shown by In islands self-assembled on both on-axis and vicinal (111)A substrates (green stars and red circles, respectively, in Figure 2), thus excluding an origin related to the presence of the strong anisotropy in the adatom diffusion coefficient on vicinal substrates due to Ehrlich–Schwöbel barrier at the step edges [20]. The temperature error bar in our measurements is associated with the accuracy of the substrate temperature determination by the thermocouple and equals approximately ±5 °C. The temperature or density error bar is not presented, if it is less than the size of red circles or green stars.

Data in Figure 2 were conveniently analyzed by considering separately liquid and solid droplets. The RHEED diffraction pattern was monitored during the deposition to assess the indium island state. A halo pattern was observed at deposition temperatures of 160 and 185 °C (see Figure 3a), which confirms the self-assembly of liquid droplets on the surface at temperatures above the In melting temperature $T_{In}^{melt}$ ≈ 157 °C. On the contrary, at temperatures below $T_{In}^{melt}$, a spotty pattern was observed (see Figure 3b), indicating the formation of epitaxial crystalline islands. It is worth mentioning that no indication of the formation of liquid droplets or solid islands can be obtained from RHEED at *T* > 200 °C, due to the low density of the formed islands and only (2 × 2) reconstruction of GaAs(111)A surface was observed during the deposition of indium.

A detailed analysis of the RHEED patterns recorded during the deposition of In at 80 °C (see Figure 4) permits to assess the crystal structure and the lattice parameter of the indium crystalline islands. Before the indium deposition, only (2 × 2)-GaAs(111)A reconstruction was observed (Figure 4a). After the deposition of 2 ML In, the diffraction reflexes of (2 × 2) reconstruction disappeared and (1 × 1)-GaAs(111)A reconstruction and additional spotty and elongated reflexes were observed in $<1\;\bar{1}\;0>$ and $<\bar{2}\;1\;1>$ azimuths (Figure 4c–f). The appearance of the additional spots shows that the indium islands are crystalline, but the growth is not pseudomorphic to the GaAs(111)A substrate. The calculated interplanar distances of In islands along each of the equivalent direction on (111) surface are the same ($l_{1}$ = 0.336 ± 0.002 nm along $<1\;\bar{1}\;0>$ and $l_{2}$ = 0.584 ± 0.005 nm along $<\bar{2}\;1\;1>$). Additionally, the ratio $l_{2}$/$l_{1}$ equals 1.74 ± 0.03 agrees with value $\sqrt{3}$ for a cubic crystal. This observation confirms that In islands, grown on vicinal GaAs(111)A substrate at 80 °C, have FCC crystal structure with lattice constant $a_{In}^{FCC}$ = 0.475 ± 0.003 nm.

Figure 4b shows RHEED intensity dependence of In and GaAs reflexes taken along the $\left[0\;\bar{1}\;1\right]$ azimuth. The points where the intensity was measured are pointed by arrows on Figure 4a,c. The intensity of the In reflex at 0 ML is not zero because there is a diffusive background from GaAs surface reconstruction. The In reflex intensity decreasing up to ~0.5 ML is related to drastic decreasing of GaAs reflex intensity and its diffusive background. After 0.5 ML, the In reflex intensity starts increasing while GaAs reflex decreases until the end of the 2 ML deposition. It is related to a nearly pseudomorphic state of initially small islands and/or to low sensitivity of RHEED technique.

Bulk indium has body-centered tetragonal (BCT) lattice. Any BCT lattice can be also represented as face-centered tetragonal (FCT) lattice, which, in the case of In has the following parameters; $a^{\prime }$ = 0.460 nm and $c^{\prime }$ = 0.495 nm at 300 K [29]. Considering thermal expansion [30], the FCT lattice constants at 80 °C (353.15 K) are $a^{\prime }$ = 0.466 nm and $c^{\prime }$ = 0.500 nm. A slight expansion of $a^{\prime }$-axis of 1.8% and a compression of $c^{\prime }$-axis by 5% results in transition from FCT to FCC lattice in agreement with our observation. FCC lattice of indium have been already observed for In nanoparticles (NPs) [29,31] and indium islands [32] with a lattice constant of 0.47–0.50 nm [29,31,32]. It is assumed that BCT–FCC transition occurs because of the surface tension with little volume dilatation at a small size of NPs. Thus, indium islands deposited on vicinal GaAs(111)A substrate at T < $T_{In}^{melt}$ are relaxed, with lattice constant that closely matches that of In FCC crystal.

Moreover, it is necessary to emphasize that BCT lattice of indium has a lattice mismatch with GaAs of ~19%. It means that, evidently, the In/GaAs interface is such that a large portion of the big mismatch is accommodated by plastic deformation. Moreover, there is a residual strain, due to BCT–FCC transition.

Our observations points that $T_{In}^{melt}$ splits the island density temperature dependence into two zones (separated by a vertical orange line in Figure 2), namely, above $T_{In}^{melt}$, where the islands are liquid indium droplets, and below $T_{In}^{melt}$, where the islands are slightly strained, indium FCC nanocrystals. In the liquid phase, the indium island density is increasing with decreasing the temperature. Despite the behavior being simple, it is worth noting that the indium island density deviates from the single exponential law predicted by straightforward application of nucleation theory [22,23], showing reduced activation energy at T < 300 °C with respect to the higher temperature values. This observation is consistent with other studies regarding Ga droplets on GaAs(001) [24], GaAs(111)A [26] substrates, and on SiN${}_{x}$ membranes [33]. Such behavior is attributed to the diffusive movement of small droplets clusters which may contribute to the subsequent coalescence and to ripening of small metal clusters, which reduce the number of islands on the surface [24,26,33].

As the temperature is lowered and solidification takes place, the density dependence on temperature completely changes behavior. At first, a surprising decrease with decreasing temperature is observed until, at even lower temperatures (~80 °C), another drastic change in slope takes place bringing back the temperature dependence of the island density to the expected increase with decreasing temperature.

At sufficiently low temperature the density of islands grows with decreasing temperature (extreme right portion of Figure 2) and it is a direct consequence of reduced diffusion length of In adatoms [24,25,26]. The island density, after reaching a minimum approximately 80 °C, increases by increasing the temperature until $T_{In}^{melt}$. Interpreting such behavior in the island density vs. temperature dependence using standard nucleation theory is impossible. Within this approach possible sources of the activation energy change (the slope in the log(island density) vs. 1/*T*) could be linked to an increase/decrease of the critical nucleus size or to a non-monotonic change of the adatom diffusivity, which could suffer below the In solidification temperature [22,23]. Both explanations are, however, hardly justifiable as the change of the critical nucleus is not expected and no transition in surface reconstruction of the GaAs surface is observed at $T_{In}^{melt}$ to explain a different adatom diffusivity. Nevertheless, within the nucleation theory approach [22,23], the activation energy can change the value but cannot change the sign and become negative.

Here, we point out that the observed behavior could be justified by the onset of a coarsening process, active when the islands are solid and not when they are in the liquid phase. Large mass transfer between islands would reduce the measured island density, with respect to the one expected based on critical nucleus size and diffusivity. A fingerprint of its presence can be gained by the analysis of the size distribution of islands (see Figure 5), as a sizable increase of the distribution is expected in the case of active coarsening phenomena. In fact, the island ensemble distribution clearly broadens, with the full width at half maximum (FWHM) of normalized (to mean size) lateral size distribution of In islands, passing from 0.8 at 160 °C (Figure 5a) to 1.5 at 30 °C (Figure 5c). It is worth mentioning that a broadening of the island size could originate from a reduction of the critical nucleus size, but it should be accompanied by a concurrent increase of the island density [34]. As island coarsening is intrinsically a kinetic effect, as originating from island to island mass transfer, to provide and additional proof of its presence we grew a sample (see Figure 1f) at the same temperature and flux of the one (see Figure 1e) resulting in the maximum deviation (T = 80 °C), but reduced the deposition time by one half (brown square in Figure 2). The island density increases, thus showing that the reentrant behavior of the island density is caused by a kinetically controlled coarsening effect.

The presence of an effective mass transfer when the islands are in the solid phase and not in the liquid phase could be traced back to the residual strain in the In crystalline islands. Strain and its local relaxation are powerful physical phenomena which control the interaction between neighboring islands, thus affecting the self-assembly island dynamics and statistics [35]. Above $T_{In}^{melt}$, the islands are constituted by indium droplets. The liquid state of the droplets makes them able to accommodate any strain. Therefore, it is reasonable to not expect any change in the island density and statistical distribution within the droplet ensemble related to the presence of strain. A completely different scenario is expected when the islands are crystalline. In the crystal phase, the strain can be accommodated via an enhancement of the height to base ratio of the islands, which allows for strain relaxation [36,37,38]. As the volume of the island increases, a large height to base ratio allows for a stronger strain relaxation. However, it also requires a large cost in terms of surface energy [35]. Therefore, in an island growing in volume, after a critical size is reached, insertion of a dislocation within the island lowers the need for strain relaxation [38], because the dislocation induced strain relaxation strongly reduces the chemical potential of the dislocated crystalline islands with respect to the islands without a new additional dislocation. Moreover, coarsening in the island ensemble is expected [39]. The dislocated island increases in volume at the expenses of the neighboring islands lying within an indium adatom diffusion length. This phenomenon was reported in several experiments on island formation in the presence of strain (see, e.g., Figure 1 in [40]). The net effect of the strain reduction by dislocation insertion, observed in the solid In islands, should be then the onset of a strong mass transfer effect that results in the reduction of the island density as the island volume exceeds the critical volume, with respect to the purely pseudomorphic islands case, as those islands which are close to a dislocated one disappear due to the strong mass transfer. As discussed in the analysis of the RHEED patterns, we have evidence that the solid In islands are not pseudomorphic, with a nearly completed strain relaxation. Thus, supporting the interpretation of an origin of the coarsening effect as due to dislocation induced changes in the chemical potential of the islands.

## 4. Conclusions

We have shown that In islands deposited on GaAs (111)A substrates, both nominal and vicinal with 2° miscut towards $\left(\bar{1}\;\bar{1}\;2\right)$, display a complex non-monotonic dependence in terms of density vs. temperature. The usual behavior, well described by nucleation theory [22,23], is maintained only until the islands remain in the liquid phase (above $T_{In}^{melt}$ ~160 °C). When the islands crystallize, coarsening phenomena take place, leading to broadening of the island distribution and to a reduction in the island density. At lower temperature (below 80 °C), islands density increases again is due to the reduction of the diffusion length which limits the range of mass transfer phenomena. The origin of the coarsening effect active only in the solid phase could be related to the presence of strain and strain relaxation in the epitaxial In islands which activates a strong mass transfer between islands. From RHEED observations, indium solid islands grown at 80 °C have FCC structure with lattice constant $a_{In}^{FCC}$ = 0.475 ± 0.003 nm. 

## Figures and Tables

**Figure 1 nanomaterials-10-01512-f001:**
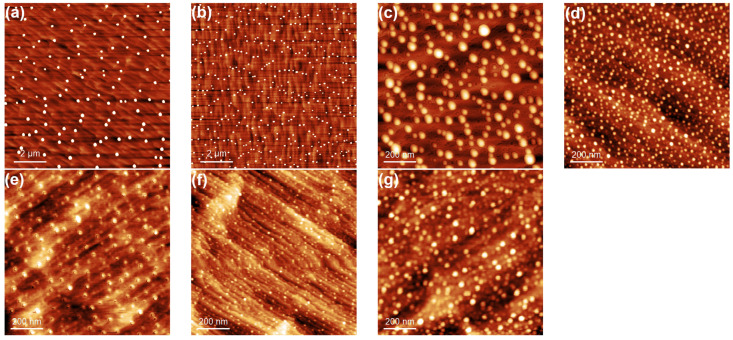
Atomic force microscope (AFM) images of indium islands, obtained by depositing 2 ML of In on GaAs(111)A with 2° miscut towards $\left(\bar{1}\;\bar{1}\;2\right)$ at (**a**) 395 °C (10 × 10 μm${}^{2}$), (**b**) 370 °C (10 × 10 μm${}^{2}$), (**c**) 240 °C (1 × 1 μm${}^{2}$), (**d**) 160 °C (1 × 1 μm${}^{2}$), (**e**) 80 °C (1 × 1 μm${}^{2}$), and (**g**) 30 °C (1 × 1 μm${}^{2}$). (**f**) AFM image of indium islands, obtained by depositing 1 ML of In on GaAs(111)A with 2° miscut towards $\left(\bar{1}\;\bar{1}\;2\right)$ at (a) 80 °C (1 × 1 μm${}^{2}$).

**Figure 2 nanomaterials-10-01512-f002:**
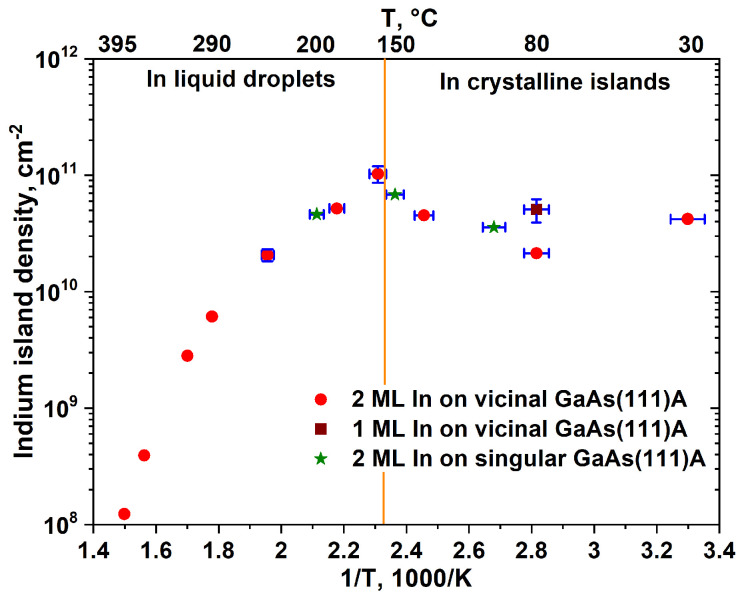
Temperature density dependence of self-assembled indium islands. Red circles correspond to 2 ML In deposited on vicinal GaAs(111)A with 2° miscut towards $\left(\bar{1}\;\bar{1}\;2\right)$; the brown square corresponds to 1 ML In deposited on vicinal GaAs(111)A with 2° miscut towards $\left(\bar{1}\;\bar{1}\;2\right)$ at 80 °C; and green stars correspond to 2 ML In deposited on singular GaAs(111)A at 100, 150, and 200 °C. Temperature error is ±5 °C. The temperature or density error bar is not presented if it is less than the size of red circles or green stars. Orange line indicates the indium melting point ($T_{In}^{melt}$ = 156.6 °C).

**Figure 3 nanomaterials-10-01512-f003:**
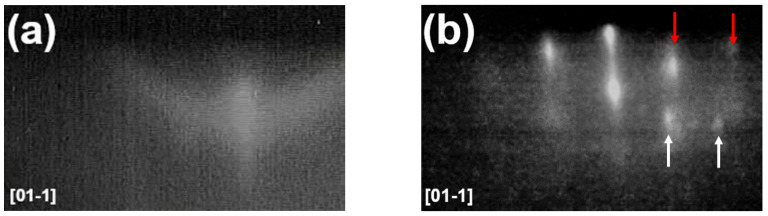
Reflection of high energy electron diffraction (RHEED) patterns of 2 ML In deposited on vicinal GaAs(111)A with 2° miscut towards $\left(\bar{1}\;\bar{1}\;2\right)$ at (**a**) 185 °C (halo pattern) and at (**b**) 80 °C (spotty pattern). Red arrows indicate RHEED reflexes from indium crystalline islands and white arrows indicate RHEED reflexes from GaAs.

**Figure 4 nanomaterials-10-01512-f004:**
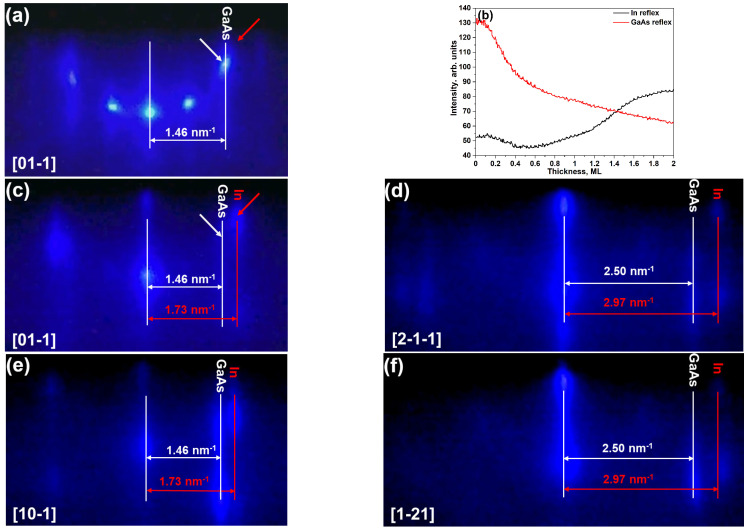
(**a**) RHEED pattern of GaAs buffer before the deposition of In. (**b**) The time dependence of the intensity of RHEED reflexes from GaAs and In taken along the $\left[0\;\bar{1}\;1\right]$ azimuth. RHEED patterns along (**c**) $\left[0\;\bar{1}\;1\right]$, (**d**) $\left[2\;\bar{1}\;\bar{1}\right]$, (**e**) $\left[1\;0\bar{1}\right]$, and (**f**) $\left[1\;\bar{2}\;1\right]$ azimuths after the deposition of 2 ML In on vicinal GaAs(111)A with 2° miscut towards $\left(\bar{1}\;\bar{1}\;2\right)$ at 80 °C. Red and white arrows correspond to points where the intensity dependence of In and GaAs reflexes, respectively, on panel (b) was measured.

**Figure 5 nanomaterials-10-01512-f005:**
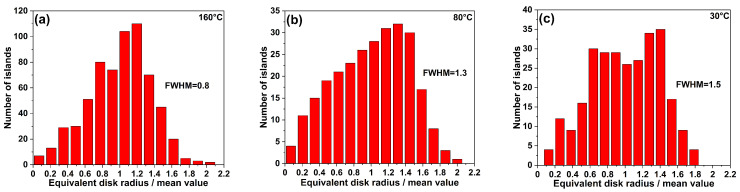
Size distributions of indium islands normalized to their mean size at (**a**) 160 °C, (**b**) 80 °C, and (**c**) 30 °C calculated from 1 × 1 μm${}^{2}$ AFM scans of each sample. The mean equivalent disk radius for panel (**a**) is 8.5 ± 3.0 nm, for panel (**b**) is 9.7 ± 4.1 nm, and for panel (**c**) is 10.6 ± 4.1 nm.

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
