# Peer review of "Reentrant Behavior of the Density vs. Temperature of Indium Islands on GaAs(111)A"

_nanomaterials, 2020, doi:10.3390/nano10081512_

Round 1
Reviewer 1 Report
- Please extend the introduction. The level of achievements (with In islands or NPs) in this area not described. The motivation of In islands application for InAs QDs growth is not presented. Please provide the advantages of InAs QDs growth via In DE versus InAs QDs growth via Stranski Krastanov.
- How the authors can relate low temperature In islands formation investigation with InAs QDs formation in future applications? Wht is the reason of low temperature investigation?
- The part Experimental methods is too compact. The mechanism of In island formation by depositing 2 ML at 80oc is not clear. The selection of different substrates (111), vicinal and singular is not motivated. Why not (100) orientation? The results of other works on GaAs (100) are not discussed and presented
- Figure 1 shows the dependence of In island density on temperature of deposition. There is no difference between 2ML depositions on singular and on vicinal substrates as well as In island density of 1ML In deposition is comparable with both 2 ML on vicinal and singular substrates
- How can You explain why the density of In nanocrystals is constant (the error bar for stars in density scale is not presented, one can be suggested from figure that density is similar; text lines 121-125 is not persuasive) in wide range of temperatures below the melting T? Please provide the growth mechanism (sketch)
- A detailed analysis of the RHEED patterns is ok. No comments
- Luck of AFM images at different deposition temperatures and ML numbers. Please update the manuscript.
Author Response
- Please extend the introduction. The level of achievements (with In islands or NPs) in this area not described. The motivation of In islands application for InAs QDs growth is not presented. Please provide the advantages of InAs QDs growth via In DE versus InAs QDs growth via Stranski Krastanov.
Thanks to the reviewer for the comments. In the introduction of the manuscript there is a lack of description of DE advantages compare to SK method. We added several paragraphs about it.
SK is strain-driven growth technique. Therefore, it is impossible to obtain QDs lattice-matched to the substrate (like GaAs/AlGaAs QDs). But DE allow to fabricate such kind of dots. Nevertheless, the main advantage of DE compare to SK growth mode is the possibility of independent control of QD density and the size of QD by controlling the deposition temperature and the amount of group III material deposited.
Additionally, SK dots cannot be achieved on (111) surfaces, while using DE it is possible to obtain highly symmetric QDs on (111) substrates, which can be used for entangled photon emission.
- How the authors can relate low temperature In islands formation investigation with InAs QDs formation in future applications? Wht is the reason of low temperature investigation?
Using DE, the density of QDs can be controlled in the wide range by the temperature of the droplet deposition. Decreasing the deposition temperature, the droplet density increases. So, if we need high QD density (laser, photodetectors), it is necessary to use low deposition temperature for droplet nucleation.
It was added several sentences and citations about the applications of high-density DE QDs.
- The part Experimental methods is too compact. The mechanism of In island formation by depositing 2 ML at 80oc is not clear. The selection of different substrates (111), vicinal and singular is not motivated. Why not (100) orientation? The results of other works on GaAs (100) are not discussed and presented.
Our interest is the fabrication of telecom QDs for entangled photon emission. One of the main problems is fine structure splitting (FSS) of the exciton state, due to QD anisotropies (shape, composition etc.), generates a decoherence mechanism which complicates the observation of the entanglement. The growth on (111) surface allow to obtain highly symmetric QDs with low FSS. But there is an additional problem, related to the growth conditions on singular GaAs(111)A substrates. To obtain very flat surface it is necessary to grow with very low growth rate and high V/III ratio. But if we use slightly miscuted (111) substrates, step-flow growth mode helps to grow thick layers with smooth surface using standard growth conditions as for the growth on GaAs(001). During the study of In islands nucleation on miscuted GaAs(111)A substrates we observed the unexpected reentrant behavior of temperature density dependence. We have grown additional samples on singular (111)A substrates, just to check that there is no connection with miscut, because before we observed a change in the activation energy for Ga droplet nucleation on miscuted GaAs(111)A substrates [A].
It is very difficult to compare with (001) surface, due to the fact that before start to nucleate any droplets it is necessary to deposit at least 1 ML of In on GaAs(001) surface to obtain metallic-rich surface. It means that the surface became InAs and In adatom surface diffusion on InAs is very huge, so the effect of reentrant behavior is very difficult to observe due to low density of islands even at very low deposition temperature. In particular it was not observed while In island and InAs QD fabrication on InP(001) [B]. And there is no any work about the nucleation of In islands or DE InAs QDs at temperatures below In melting point.
[A] Tuktamyshev, A.; Fedorov, A.; Bietti, S.; Tsukamoto, S.; Sanguinetti, S. Temperature Activated Dimensionality Crossover in the Nucleation of Quantum Dots by Droplet Epitaxy on GaAs(111)A Vicinal Substrates. Sci. Rep. 2019, 9, 14520. https://doi.org/10.1038/s41598-019-51161-5.
[B] Fuster, D.; Abderrafi, K.; Alén, B.; González, Y.; Wewior, L.; González, L. InAs Nanostructures Grown by Droplet Epitaxy Directly on InP(001) Substrates. J. Cryst. Growth 2016, 434, 81-–87. https://doi.org/10.1016/j.jcrysgro.2015.11.003.
- Figure 1 shows the dependence of In island density on temperature of deposition. There is no difference between 2ML depositions on singular and on vicinal substrates as well as In island density of 1ML In deposition is comparable with both 2 ML on vicinal and singular substrates
For both nominal and miscut GaAs(111)A substrates, only (2x2) surface reconstruction is observed. Therefore, we expect to have same critical nucleus size for In droplets and same diffusion behavior for In adatoms. Thus, the density dependence of In islands on both GaAs(111)A substrates is identical. As we expected to have less In island density at less deposited material for temperatures below the melting point, we have grown additional sample at 80 °Ð¡ with 1 ML coverage. This sample show the In island density of 5.1±1.2x1010 cm-2, which is almost 2.5 times more that for deposition of 2ML In at the same temperature (2.14±0.02x1010 cm-2).
- How can You explain why the density of In nanocrystals is constant (the error bar for stars in density scale is not presented, one can be suggested from figure that density is similar; text lines 121-125 is not persuasive) in wide range of temperatures below the melting T? Please provide the growth mechanism (sketch)
The density of In islands below the In melting point is not a constant as can be seen from the figure below. And we did not present the error bar if the error less than the size of red circle or green star. Now we mentioned this fact in the text and figure caption.
From the nucleation theory, the density of droplets depends only on the deposition temperature and flux. In the beginning of deposition all droplets are nucleated and with increase the coverage, only the size of droplets increases. In case of deposition below In melting point, there is a strong mass transfer between islands due to different number of misfit dislocation in the In-GaAs interface, which neglect below approximately 80 C due to the short diffusion length of In adatoms and the density of In islands again increases with decreasing the deposition temperature.
- A detailed analysis of the RHEED patterns is ok. No comments
- Luck of AFM images at different deposition temperatures and ML numbers. Please update the manuscript.
We added AFM images of In islands obtained on the vicinal GaAs(111)A substrates at different deposition temperatures.

Reviewer 2 Report
This is a very interesting experimental study, which gives an insight into the initial stages of the droplet epitaxy. I recommend accepting it for publication. I have just minor remarks.
- Page 1, line 18, “…One possible way is to fabricate InAs QDs embedded in an InGa(Al)As barrier, metamorphically grown on GaAs substrates [7–9].”
The above statement is inaccurate.
Ref. 7 did use GaAs substrates but they did not use an InGaAlAs barrier metamorphically grown on GaAs substrates. BTW, the InAs QDs were Stranski-Krastanow ones (i.e. no droplet epitaxy there). BTW2, the arXiv paper [7] is now published in Communications Physics, https://doi.org/10.1038/s42005-020-0390-7 .
Ref. 9 did not use GaAs substrates. They used InP (111)A substrates, and InGaAlAs was lattice matched to InP.
Therefore, I propose to modify the above statement in the manuscript accordingly. In addition, as you cited a paper on InAs droplet QDs grown on InP substrates, it is worth adding another citation: J. Skiba-Szymanska, R. M. Stevenson, C. Varnava, M. Felle, J. Huwer, T. Müller, A.J. Bennett, J.P. Lee, I. Farrer, A.B. Krysa, P. Spencer, L.E. Goff, D.A. Ritchie, A.J. Shields, “Universal Growth Scheme for Quantum Dots with Low Fine-Structure Splitting at Various Emission Wavelengths”, Physical Review Applied, Volume 8, Issue 1, 2017, Article number 014013, https://doi-org.sheffield.idm.oclc.org/10.1103/PhysRevApplied.8.014013
The latter study presented InAs droplet QDs grown on InP emitting around 1550 nm with a small FSS suitable for the fabrication of entangled LEDs.
Other good references for InAs QD epitaxy on metamorphic buffers on GaAs would be:
Metamorphic growth for application in long-wavelength (1.3–1.55 µm) lasers and MODFET-type structures on GaAs substrates, Semenova, E S; Zhukov, A E; Mikhrin, S S; Egorov, A Yu; Odnoblyudov, V A; Vasil ev, A P; Nikitina, E V; Kovsh, A R; Kryzhanovskaya, N V; Gladyshev, A G; Blokhin, S A; Musikhin, Yu G; Maximov, M V; Shernyakov, Yu M; Ustinov, V M; Ledentsov, N N, DOI: 10.1088/0957-4484/15/4/031, Nanotechnology , 2004, Vol.15(4), p.S283-S287
Metamorphic approach to single quantum dot emission at 1.55μm on GaAs substrate, Semenova, E S; Hostein, R; Patriarche, G; Mauguin, O; Largeau, L; Robert-Philip, I; Beveratos, A; Lemaître, A, DOI: 10.1063/1.2927496, Journal of applied physics. , 2008, Vol.103(10), p.103533
InAs quantum dots grown on metamorphic buffers as non-classical light sources at telecom C-band: a review, Portalupi, Simone Luca; Jetter, Michael; Michler, Peter, DOI: 10.1088/1361-6641/ab08b4, Semiconductor science and technology , 2019, Vol.34(5), p.053001
- Page 2, line 32, “…miscuted…”
…miscut…
- Page 2, line 40, “A 130 nm GaAs buffer layer was deposited, to smooth the surface.”
The above statement suggests that the surface of the substrate was rough. It was probably rough after the deoxidation process. However, the latter step was not mentioned.
- Does the size of the In droplets affect the melting point?
- The manuscript requires some minor language corrections, e.g. “A 130 nm GaAs buffer layer was deposited, to smooth the surface.” I would suggest add “thick” and delete the comma: “A 130nm thick GaAs buffer layer was deposited to smooth the surface”.
- It is worth mentioning that an appearance of diffraction reflexes from In islands occurs after the deposition of 0.5 ML of indium (see Figure3b), which can be related to a nearly pseudomorphic state of initially small islands and/or to low sensitivity of RHEED technique.
Fig 3b requires some further clarification. What azimuth is associated with Fig. 3b? Even at 0 ML (i.e. no In) the intensity of the reflex from In is above zero, ~53 a.u. This can be confusing for the readers who are not familiar with RHEED from In droplets. Why is it above zero if there is no In? Due to the superposition of In and GaAs reflexes? The intensity of the In reflex drops from ~54 a.u. (0.1ML) to a plateau ~45 a.u. (0.4-0.6 ML). Then, it increases at a rate of ~20 a.u./ML within 0.7-1.2 ML. Then, within 1.2-1.4 ML, the intensity increases even at a higher rate. The intensity saturates at ~1.9ML. Is this saturation expected?
- 2b and Fig 3c depict RHEED, which is recorded at the same conditions (2ML of In, 80oC, [01-1] azimuth, 2deg miscut). Therefore, one would expect these Figures to be identical. However, they are not identical. Why?
- Page 5, line 146. “As island coarsening is intrinsically a kinetic effect, as originating from island to island mass transfer, to provide and additional proof of its presence we grew a sample at the same temperature and flux of the one resulting in the maximum deviation (T = 80 C), and reduced the deposition time by one half (brown square in Figure1).”
There is a typo (and->an), I believe. As an option, I would propose to rephrase the sentence in the following way:
As island coarsening is intrinsically a kinetic effect, originating from island to island mass transfer, in order to provide an additional proof of its presence we grew a sample at the same temperature and flux of the one resulting in the maximum deviation (T = 80 C), but reduced the deposition time by one half (brown square in Figure1).
However, please, check whether my suggestion does not change the original meaning.
What is "the maximum deviation"? The deviation of what?
Author Response
This is a very interesting experimental study, which gives an insight into the initial stages of the droplet epitaxy. I recommend accepting it for publication. I have just minor remarks.
1 Page 1, line 18, “…One possible way is to fabricate InAs QDs embedded in an InGa(Al)As barrier, metamorphically grown on GaAs substrates [7–9].”
The above statement is inaccurate.
Ref. 7 did use GaAs substrates, but they did not use an InGaAlAs barrier metamorphically grown on GaAs substrates. BTW, the InAs QDs were Stranski-Krastanow ones (i.e. no droplet epitaxy there). BTW2, the arXiv paper [7] is now published in Communications Physics, https://doi.org/10.1038/s42005-020-0390-7.
Ref. 9 did not use GaAs substrates. They used InP (111)A substrates, and InGaAlAs was lattice matched to InP.
Therefore, I propose to modify the above statement in the manuscript accordingly. In addition, as you cited a paper on InAs droplet QDs grown on InP substrates, it is worth adding another citation: J. Skiba-Szymanska, R. M. Stevenson, C. Varnava, M. Felle, J. Huwer, T. Müller, A.J. Bennett, J.P. Lee, I. Farrer, A.B. Krysa, P. Spencer, L.E. Goff, D.A. Ritchie, A.J. Shields, “Universal Growth Scheme for Quantum Dots with Low Fine-Structure Splitting at Various Emission Wavelengths”, Physical Review Applied, Volume 8, Issue 1, 2017, Article number 014013, https://doi-org.sheffield.idm.oclc.org/10.1103/PhysRevApplied.8.014013
The latter study presented InAs droplet QDs grown on InP emitting around 1550 nm with a small FSS suitable for the fabrication of entangled LEDs.
Other good references for InAs QD epitaxy on metamorphic buffers on GaAs would be:
Metamorphic growth for application in long-wavelength (1.3–1.55 µm) lasers and MODFET-type structures on GaAs substrates, Semenova, E S; Zhukov, A E; Mikhrin, S S; Egorov, A Yu; Odnoblyudov, V A; Vasil ev, A P; Nikitina, E V; Kovsh, A R; Kryzhanovskaya, N V; Gladyshev, A G; Blokhin, S A; Musikhin, Yu G; Maximov, M V; Shernyakov, Yu M; Ustinov, V M; Ledentsov, N N, DOI: 10.1088/0957-4484/15/4/031, Nanotechnology , 2004, Vol.15(4), p.S283-S287
Metamorphic approach to single quantum dot emission at 1.55μm on GaAs substrate, Semenova, E S; Hostein, R; Patriarche, G; Mauguin, O; Largeau, L; Robert-Philip, I; Beveratos, A; Lemaître, A, DOI: 10.1063/1.2927496, Journal of applied physics. , 2008, Vol.103(10), p.103533
InAs quantum dots grown on metamorphic buffers as non-classical light sources at telecom C-band: a review, Portalupi, Simone Luca; Jetter, Michael; Michler, Peter, DOI: 10.1088/1361-6641/ab08b4, Semiconductor science and technology , 2019, Vol.34(5), p.053001
We would like to thank the reviewer for the helpful comments.
In the work [7] InGaAs strained relaxed layer (SRL) was used to obtain an emission at telecom wavelengths from SK InAs QDs grown on GaAs substrate. And in the work [9] authors used InGaAlAs buffer layer pseudomorphically grown on InP substrate. Therefore, the sentence “One possible way is to fabricate InAs QDs embedded in an InGa(Al)As barrier, metamorphically grown on GaAs substrates [7–9].” was corrected by “One possible way is to fabricate InAs QDs embedded in InGa(Al)As layers, pseudomorphically grown on InP substrates [a, b] or metamorphically grown on GaAs substrates [c, d, e, f].
[a] Skiba-Szymanska, J.; Stevenson, R. M.; Varnava, C.; Felle, M.; Huwer, J.; Müller, T.; Bennet, A. J.; Lee, J. P.; Farrer, I.; Krysa, A. B.; Spencer, P.; Goff, L. E.; Ritchie, D. A.; Heffernan, J.; Shields A. J. Universal growth scheme for quantum dots with low fine-structure splitting at various emission wavelength. Phys. Rev. Appl. 2017, 8, 014013. https://doi.org/10.1103/PhysRevApplied.8.014013
[b] Ha, N.; Mano, T.; Dubos, S.; Kuroda, T.; Sakuma, Y.; Sakoda, K. Single Photon Emission from Droplet Epitaxial Quantum Dots in the Standard Telecom Window around a Wavelength of 1.55 mm. Appl. Phys. Express 2020, 13, 025002. https://doi.org/10.35848/1882-0786/ab6e0f.
[c] Semenova, E. S.; Zhukov, A. E.; Mikhrin, S. S.; Egorov, A. Yu.; Odnoblyudov, V. A.; Vasil’ev, A. P.; Nikitina, E. V.; Kovsh, A. R.; Kryzhanovskaya, N. V.; Gladyshev, A. G.; Blokhin, S. A.; Musikhin, Yu.G.; Maximov, M. V.; Shernyakov, Yu.M.; Ustinov, V. M.; Ledentsov, N. N. Metamorphic growth for application in long-wavelength (1.3–1.55 μm) lasers and MODFET-type structures on GaAs substrates. Nanotechnology 2004, 15, S283 – S287. https://doi.org/10.1088/0957-4484/15/4/031
[d] Semenova, E. S.; Hostein, R.; Patriarche, G.; Mauguin, O.; Largeau, L.; Robert-Philip, I. Beveratos, A.; Lemaître, A. Metamorphic approach to single quantum dot emission at 1.55 μm on GaAs substrate. J. Appl. Phys. 2008, 103, 103533. https://doi.org/10.1063/1.2927496
[e] Portalupi, S. L.; Jetter, M.; Michler, P. InAs quantum dots grown on metamorphic buffers as non-classical light sources at telecom C-band: a review. Semicond. Sci. Technol. 2019, 34, 053001. https://doi.org/10.1088/1361-6641/ab08b4.
[f] Zeuner, K. D.; Jöns, K. D.; Schweickert, L.; Hedlund, C. R.; Lobato, C. N.; Lettner, T.; Wang, K.; Gyger, S.; Schöll, E.; Steinhauer, S.; Hammar, M.; Zwiller, V. On-Demand Generation of Entangled Photon Pairs in the Telecom C-Band for Fiber-Based Quantum Networks. arXiv:1912.04782 2019. https://arxiv.org/abs/1912.04782.
2 Page 2, line 32, “…miscuted…”
…miscut…
The “…miscuted…” was corrected to “…miscut…”.
3 Page 2, line 40, “A 130 nm GaAs buffer layer was deposited, to smooth the surface.”
The above statement suggests that the surface of the substrate was rough. It was probably rough after the deoxidation process. However, the latter step was not mentioned.
Yes, the GaAs surface is rough after thermal desorption of an oxide and it is necessary to deposit buffer layer to smooth the surface. Therefore, the sentence “A 130 nm thick GaAs buffer layer was deposited, to smooth the surface.” was corrected by “After an oxide desorption at 590 °Ð¡ for 5 minutes a 130 nm GaAs buffer layer was deposited to smooth the surface”.
4 Does the size of the In droplets affect the melting point?
From our knowledge the size of In droplets doesn’t affect the In melting point. Probably this question raised from the not clear sentences in lines 128-133 on the page 5: “The island density, after reaching a minimum around 80 °Ð¡, increases by increasing the temperature until Tmelt(In). Interpreting such change in the island density vs temperature dependence using standard nucleation theory is not simple. Within this approach possible sources of the observed behavior could be linked to an increase of the critical nucleus size, lowering the temperature, when the In phase changes from liquid to solid or to a non-monotonic change of the adatom diffusivity, which could suffer from a drastic increase below the In solidification temperature [13,14].”
From the classical nucleation theory [13, 14] the activation energy can be expressed as follows:
,
where Ed is the diffusion energy, i is critical nucleus size, Ei – binding energy for critical cluster (the energy needed to decompose critical cluster) and p = i/(i+2) for 2D diffusion behavior. From this expression, it is clear that Ea cannot be negative, only can change the value if critical cluster size or diffusivity are changed.
So, above mentioned sentences was modified as “The island density, after reaching a minimum around 80 â—¦C, increases by increasing the temperature until Tmelt(In). Interpreting such behavior in the island density vs temperature dependence using standard nucleation theory is immposible. Within this approach possible sources of the activation energy change (the slope in the log(island density) vs 1/T) could be linked to an increase/decrease of the critical nucleus size or to a non-monotonic change of the adatom diffusivity, which could suffer below the In solidification temperature [22,23]. Both explanations are, however, hardly justifiable as the change of the critical nucleus is not expected and no transition in surface reconstruction of the GaAs surface is observed at Tmelt(In) to explain a different adatom diffusivity. Anyway, within the nucleation theory approach [22,23] the activation energy can change the value but cannot change the sign and became negative.”.
5 The manuscript requires some minor language corrections, e.g. “A 130 nm GaAs buffer layer was deposited, to smooth the surface.” I would suggest add “thick” and delete the comma: “A 130nm thick GaAs buffer layer was deposited to smooth the surface”.
The same as for comment 3: the sentence “A 130 nm thick GaAs buffer layer was deposited, to smooth the surface.” was corrected by “After an oxide desorption at 590 °Ð¡ for 5 minutes a 130 nm GaAs buffer layer was deposited to smooth the surface”.
6 It is worth mentioning that an appearance of diffraction reflexes from In islands occurs after the deposition of 0.5 ML of indium (see Figure3b), which can be related to a nearly pseudomorphic state of initially small islands and/or to low sensitivity of RHEED technique.
Fig 3b requires some further clarification. What azimuth is associated with Fig. 3b? Even at 0 ML (i.e. no In) the intensity of the reflex from In is above zero, ~53 a.u. This can be confusing for the readers who are not familiar with RHEED from In droplets. Why is it above zero if there is no In? Due to the superposition of In and GaAs reflexes? The intensity of the In reflex drops from ~54 a.u. (0.1ML) to a plateau ~45 a.u. (0.4-0.6 ML). Then, it increases at a rate of ~20 a.u./ML within 0.7-1.2 ML. Then, within 1.2-1.4 ML, the intensity increases even at a higher rate. The intensity saturates at ~1.9ML. Is this saturation expected?
The RHEED reflex intensity dependence taken along the [0-11] azimuth. The points where the intensity was measured are pointed by arrows on Figures 3a, 3c. The behavior of In reflex intensity dependence below the coverage of ~ 0.5 and above ~ 1.6 ML can be explained.
The intensity of In reflex at 0 ML is not zero because there is a diffusive background from GaAs surface reconstruction. The intensity decreasing up to ~ 0.5 ML is related to drastic decreasing of GaAs reflex intensity and its diffusive background. After 0.5 ML the In reflex intensity is starting increasing while GaAs reflex is decreasing until the end of the 2 ML deposition. It means that only from this point we can approve an appearance of RHEED reflex from In dislocated islands, which are not pseudomorphic to GaAs surface. Thus, before 0.5 ML coverage or In islands are pseudomorphic to GaAs substrate (which is difficult to believe due to large lattice mismatch of ~19% between bulk In crystal and GaAs), or just the RHEED technique sensitivity is not enough to see nucleation of In dislocated islands in the beginning of In deposition.
The In reflex intensity saturation after ~ 1.6 ML coverage corresponds to the fact that with increasing the volume of In island at constant In deposition rate, the island size increment decreases and saturates.
The behavior of the intensity in the range of 0.5 – 1.6 ML is difficult to explain without additional experiments. The growth of the intensity could be related to an increase of island density or an increase of island size. But this is not an interest of the given manuscript.
7 2b and Fig 3c depict RHEED, which is recorded at the same conditions (2ML of In, 80oC, [01-1] azimuth, 2deg miscut). Therefore, one would expect these Figures to be identical. However, they are not identical. Why?
Figures 2b and 3c are the same. Just for Figures 2a and 2b we used grayscale images, to show better a hallo pattern and spotty pattern due to better contrast. Now we also pointed by arrows a 3D reflexes from In islands in the Figure 2b.
For Figure 3 we used original color images to show better the positions of GaAs and In reflexes due to less background noise of colorful image. Also, Figure 3c is slightly shorter than 2b and (02) reflex for In is not presented as for Figure 2b (the second right red arrow).
8 Page 5, line 146. “As island coarsening is intrinsically a kinetic effect, as originating from island to island mass transfer, to provide and additional proof of its presence we grew a sample at the same temperature and flux of the one resulting in the maximum deviation (T = 80 C), and reduced the deposition time by one half (brown square in Figure1).”
There is a typo (and->an), I believe. As an option, I would propose to rephrase the sentence in the following way:
As island coarsening is intrinsically a kinetic effect, originating from island to island mass transfer, in order to provide an additional proof of its presence we grew a sample at the same temperature and flux of the one resulting in the maximum deviation (T = 80 C), but reduced the deposition time by one half (brown square in Figure1).
However, please, check whether my suggestion does not change the original meaning.
What is "the maximum deviation"? The deviation of what?
“The maximum deviation” means the deviation of the temperature density dependence of In islands from the monotonic increase of density with decreasing the temperature predicted by nucleation theory. At deposition temperature less than 80 °Ð¡ we again observe an increase of the island density with decreasing the deposition temperature.
We agree with the rephrasing of the sentence.

Round 2
Reviewer 1 Report
minor spell check required